# Experimental Study on Mechanical Properties of Basalt Fiber Concrete after Cryogenic Freeze−Thaw Cycles

**DOI:** 10.3390/polym15010196

**Published:** 2022-12-30

**Authors:** Yang Li, Zhicong Gu, Ben Zhao, Jiangkun Zhang, Xu Zou

**Affiliations:** 1Department of Civil Engineering, Hubei University of Technology, Wuhan 430068, China; 2Department of Civil Engineering, University of Padova, Via F. Marzolo 9, 35131 Padova, Italy

**Keywords:** basalt fiber, cryogenic cycles, compressive strength, toughness, splitting tensile strength

## Abstract

Basalt fiber (BF) has received much attention in recent years for engineering practice and scientific research related to basalt fiber reinforced concrete (BFRC) due to its advantageous mechanical properties and cost-effectiveness. By researching its performance characteristics after cryogenic freeze–thaw cycles, the advantages of BFRC’s mechanical properties can be further exploited in order to expand its application scope. The effects of the fiber volume fraction, temperature gradient, and number of freeze–thaw cycles on the compressive strength, toughness index, splitting tensile strength, flexural strength, etc., of BFRC were investigated. Additionally, the damage mechanism of BFRC after freeze–thaw cycles was analyzed via scanning electron microscopy (SEM). The results show that the compressive strength of BFRC reaches its peak value when the fraction reaches 0.1% under the conditions of freezing and thawing cycles from room temperature to −80 °C. When the fraction of BFRC is 0.1%, and the maximum reduction is 17.1%, the splitting tensile strength decreased most sharply when the fraction was 0.1%, and the decrease amplitude was 40.9%, and the flexural strength decreased most acutely when the fraction was 0.3%, and the maximum decrease was 44.62%. The addition of basalt fibers can reduce the damage to the microstructure of concrete and improve its plastic degradation characteristics to a certain extent. With a decrease in the minimum temperature of the cryogenic freeze–thaw cycle, the optimal fiber content for compressive strength increases. Nevertheless, the splitting tensile strength and flexural strength of BFRC is improved as the fiber content increases under the cryogenic freeze–thaw environment.

## 1. Introduction

In the field of civil engineering, concrete is still the most widely used construction material. Approximately 10 billion tons of concrete are produced globally every year [1]. Meanwhile, liquefied natural gas (LNG) is gradually being discovered and developed as a major energy source [2,3]. In LNG storage facilities, LNG storage tanks with concrete as the inner tank have a more prominent economic advantage compared to traditional LNG storage facilities that use 9% nickel steel as the inner tank [4]. Although using prestressed concrete as the primary LNG storage tank has great advantages, the leakage of LNG through cracks in the concrete storage tank has the potential to cause large explosions. It is essential to ensure that the LNG concrete has strong mechanical properties such as compressive strength and crack resistance. Previous studies have shown that cryogenic freeze–thaw cycles induced by LNG being transported into and out of the tank cause more significant damage to the concrete structure than the constant ultra-low-temperature environment [5,6,7]. There is an urgent need to improve the mechanical properties and crack resistance of concrete after experiencing cryogenic freeze–thaw cycles. At present, a popular method of improving the crack resistance of concrete is to add fiber [8,9]. In this regard, the excellent mechanical properties of BFRC at room temperature have been verified, while basalt fibers have strong application potential, a relatively low cost, a simple fabrication process [10], good heat resistance, corrosion resistance, low thermal conductivity [11], and so on. Therefore, it is necessary to further investigate its mechanical properties of BFRC after experiencing low- and ultra-low-temperature freezing and thawing to provide a reference for its optimal design and theoretical research in related fields.

In recent years, much research has been carried out on fiber concrete. Carvalho et al. [12] described the development of natural fibers and future trends, and the main issue they highlighted is how to obtain the appropriate fiber admixture to obtain excellent mechanical properties. Basalt fiber made from volcanic rocks after the melting process can be mixed into concrete [13]. The production process of basalt fiber is very similar to glass fiber; however, but unlike carbon fiber or glass fiber, basalt fiber has no additives, and it is cheaper. The incorporation of basalt fiber into concrete can effectively improve the energy absorption capacity and ductility properties of concrete, but it has little effect on its compressive strength [14,15]. Most studies on BFRC have focused on its basic mechanical properties: compressive strength, flexural strength, and splitting tensile strength. Studies have shown that the appropriate amount of basalt fiber is between 0.2% and 0.4% and that excessive amounts can be counterproductive [16,17]. The study of FRC damage mechanisms has also become a hot topic in recent years. Li DS et al. [18] conducted tests on polyvinyl alcohol fiber-reinforced concrete using the acoustic emission (AE) technique and obtained relevant characteristic parameter maps to clarify the damage mechanism of polyvinyl alcohol fiber-reinforced concrete. Aggelis DG et al. [19] also characterized the results of steel fiber concrete in four-point bending tests using the AE technique to reveal the damage mechanism of steel fiber concrete. Yang L et al. [20] demonstrated BFRC’s surface crack extension behavior by strain profiling with the 3D-DIC technique combined with the AE technique to monitor the internal damage. At low- and ultra-low temperatures, BFRC shows good development potential. Cheng [21] concluded that the addition of basalt fiber at low temperatures directly enhanced the crack resistance of asphalt mixes. The results of Cantin et al. [22] showed a significant increase in the toughness of steel fiber-reinforced concrete (SFRC) at low temperatures of −10 °C and −30 °C. Caballero-Jorna M et al. [23] highlighted a slight increase in flexural strength after cracking in both macroscopic synthetic fiber concrete (MSFRC) and SFRC at temperatures as low as −15 °C. However, the opposite result was obtained by Richardson A et al. [24]. They found that the flexural strength of both SFRC and MSFRC was increased at −20 °C.

The rapid growth of the LNG industry [25] has promoted the use of concrete as a storage tank infrastructure material in the ultra-low sector [26,27]. At present, cryogenic concrete has been gradually used in LNG storage tanks [28,29] with the popularization of all-concrete LNG (ACLNG) storage tanks [30,31]. The requirements for designing and constructing LNG storage tanks that meet high standards have been increasing, and the performance and ratio design of FRC under deep cooling conditions have become a hotspot of research. According to Dahmani L et al. [32], it is clear that the freezing of water within the pores of concrete in low-temperature environments can lead to concrete microcracks. Marshall A L’s study [33] pointed out that the behavior of concrete at low temperatures is mostly determined by its porosity. Rostasy F S [34,35] mentioned that after freeze–thaw cycles in concrete, the amount of cement mortar loss increases in pores larger than 50 mm, the number of freeze–thaw cycles increases continuously, and damage caused by freeze–thaw cycles will only occur in pores larger than 50 mm. In general, the existing studies mainly focus on the basic mechanical properties of concrete and its change mechanism after experiencing ultra-low temperatures, and there are relatively few studies on the mechanical properties of FRC after experiencing ultra-low temperatures, especially BFRC.

Accordingly, this study was carried out to investigate the mechanical properties of BFRC after freeze–thaw cycles at low- and ultra-low temperatures, including −80 °C, −150 °C, and −196 °C. Three different variables (fiber content, number of freeze–thaw cycles, and temperature cycling gradient) were used to investigate the compressive strength, splitting tensile strength, and flexural strength of BFRC, as well as the variation of the toughness index under uniaxial compression. In addition, scanning electron microscope (SEM) images were used to analyze the surface morphology and damage characteristics of BFRC after freeze–thaw cycles at ultra-low temperatures to reveal the mechanism of BFRC damage after cryogenic freeze–thaw cycles. The present study investigates the mechanical properties of basalt fiber-reinforced concrete in ultra-low-temperature environments and summarizes the laws surrounding them to provide effective scientific help and guidance for the application of ultra-low-temperature engineering and to further promote the development of this field.

## 2. Materials and Methods

### 2.1. Raw Materials

The cementitious material used in the test was P.O. 42.5 ordinary silicate cement produced by Huaxin Cement Co., Ltd. (Huangshi, China) according to national standard GB/T4131−2014 (“Naming Principles and Terminology of Cement”). The fine aggregate comprises medium river sand with a fineness modulus of 2.6 and a particle size of 0.35–0.5 mm according to GB/T 17617−1999 (“Test Method of Cementitious Sand Strength (ISO Method)”). The coarse aggregates were well-graded common gravels with a particle size of 5−25 mm, and the water used for the test was tap water for municipal use. The basalt fiber used in the test was short-cut basalt fiber produced by Zhejiang Haining Anjie Materials Co., Ltd. (Haining, China) A physical diagram of the basalt fiber is shown in Figure 1. The main mechanical and thermal indexes of the fiber are shown in Table 1. The morphological characterization of the fiber is shown in Figure 2.

### 2.2. Concrete Mix Design

In order to consider the concrete strength and durability in concrete design, the Chinese standards JGJ55-2011 (“Ordinary Concrete Proportioning Design Regulations”), GB/T50080-2002 (“Ordinary Concrete Mix Performance Test Methods”), JGJ/T 221−2010 (“Technical Regulations for the Application of Fiber Concrete”), and other specifications were used for reference. After several trials, the final mix ratio was determined, which is shown in Table 2. The dosage of basalt fiber was calculated according to the fiber volume rate based on the basalt fiber density value of 2640 kg/m^3^ given by the manufacturer. The mix designs of the specimen are shown in Table 3.

### 2.3. Specimen Design

In the specimen numbers: P indicates plain concrete; FB indicates basalt fiber concrete; 0, 1, 2, and 3 indicate basalt fiber volume fractions of 0.0%, 0.1%, 0.2%, and 0.3%, respectively; +20, −80, −150, and −196 indicate temperature gradients of 20 °C, 20–80 °C, 20–150 °C, and 20–196 °C, respectively; C, T, and F indicate compressive specimen, splitting tensile specimen, and flexural specimen, respectively; and 0, 3, and 6 indicate no cryogenic cycle, 3 cryogenic cycles, and 6 cryogenic cycles, respectively. For example, FB2−80C3 indicates a compressive specimen with a basalt fiber volume fraction of 0.2% experiencing 3 cryogenic cycles of 20–80 °C.

### 2.4. Test Methods

#### 2.4.1. Preparation and Maintenance of Specimens

In this test, the dry fiber mixing method was chosen to prepare the specimens. Firstly, put sand and gravel in the mixer and mix for 30 s so that the fine aggregate can be evenly dispersed among the coarse aggregate. Add cement and mix for 30 s. Then, evenly sprinkle fibers in the mixer and mix for 60 s. After the sand, stone, cement, and basalt fiber are evenly mixed, slowly add water to the mixture for no less than 60 s until it is evenly mixed.

The concrete specimens were cured by the standard curing method, and the specimens were placed in a model SHBY-40B standard constant temperature and humidity curing box. According to the standard curing requirements, the temperature of the equipment was adjusted to room temperature (20 ± 2 °C), and the relative humidity was adjusted to above 95%. The specimens were taken out after 28 days of curing.

#### 2.4.2. Cryogenic Freeze–Thaw Cycle Treatment Method

The cryogenic freeze–thaw cycle treatment of BFRC specimens was carried out in a self-developed cryogenic environment simulation chamber. The specimen temperature was evaluated by a DT80G data collector. The cryogenic chamber uses liquid nitrogen as the medium for internal vaporization to achieve the purpose of cooling. The BFRC specimens are placed inside the chamber under the temperature gradient (20–80 °C, 20–150 °C, 20–196 °C) at a cooling rate of 2 °C/min. After the specimen is lowered to the target temperature, it is necessary to maintain a constant temperature for more than 30 min to ensure that the surface temperature of the specimen and the internal core temperature are the same. Then, turn off the system and open the box, letting it return to room temperature naturally. After the specimen returned to room temperature, the above operations were repeated until all cryogenic cycles were completed.

#### 2.4.3. Mechanical Properties

The mechanical properties of BFRC were determined according to GB/T50081-2019 (“Standard for Physical and Mechanical Properties of Concrete Test Methods”). The specimens used for both the compressive strength and splitting tensile strength tests were BFRC cubes of 100 × 100 × 100 mm. The instrument used is a microcomputer-controlled hydraulic universal testing machine. The loading method of this test is displacement-controlled at a rate of 0.3 mm/min.

The pediment used for the flexural strength test is a non-standard BFRC specimen of 100 × 100 × 400 mm. The loading equipment is an elector-hydraulic servo-hydraulic flexural testing machine manufactured by Meister with a maximum load of 100 kN. The test is load-controlled at a rate of 0.6 MPa/s. The specimen size of the uniaxial compressive test is 100 × 100 × 400 mm. The main equipment used in this experiment is shown in Figure 3a,b. Part of the experimental procedure is shown in Figure 4.

## 3. Results and Discussion

### 3.1. Compressive Strength

During the loading process, the plain concrete specimen without basalt fiber was first broken at the corners. Cracks were continuously generated on the surface, and the exterior concrete began to fall off over a large area. Then, one large crack was formed through the middle. Finally, it suddenly crumbled, as is shown in Figure 5a. For the specimen with 0.1% basalt fibers, the surface penetration crack width became smaller during the loading process, and the surface did not crack extensively until complete damage, as is shown in Figure 5b. For the specimens with 0.2% and 0.3% basalt fibers, the crack initiation rate of the BFRC specimens was significantly reduced compared with the plain concrete, and no penetration cracks appeared throughout the process. The specimens remained intact when the specimens cracked, and they showed obvious plastic damage characteristics, as is shown in Figure 5c,d.

From Figure 6, it is clear that the compressive strength of BFRC reached its maximum value at 20 °C as the ambient temperature with 0.1% basalt fibers. In concrete, the appropriate content of fibers can be adequately surrounded by cement paste to form an effective three-dimensional mesh structure. When the specimens were under compression, the compressive stress was transferred to the fibers through the aggregate cement paste bond interface. Additionally, the fibers also shared the load, thus effectively improving the compressive strength. Under the freeze–thaw conditions from room temperature to −80 °C, the compressive strength increased by 8.26 MPa when the fiber volume fraction increased from 0% to 0.1%, showing a percentage improvement of about 22%. When the fiber volume fraction increased from 0.1% to 0.2% and 0.3%, the strength decreased by 3.68 MPa and 4.07 MPa, respectively, showing a percentage decrease of 8.04% and 9.66%, respectively. This indicates that excess fibers lack adequate surrounding wrapping and tend to be unevenly distributed. This finally leads to agglomeration, which in turn causes more defects in the internal structure and counteracts the effect of fibers to inhibit the development of damage, resulting in a decrease in strength. When the ultra-low temperature in the freeze–thaw environment further drops, the deterioration trend of concrete accelerates. The fiber bundles were easy to separate from the cement matrix wrapping grip due to such severe freeze–thaw cycles. However, if there is a higher amount of fiber incorporated into it, there are still sufficient fiber bundles to carry the load via connecting with the matrix and inhibiting internal cracking, and a higher amount of fiber acts as a bridge, slowing down the damage to the internal structure caused by cryogenic freeze–thaw cycles. This may explain why the volume fraction corresponding to the maximum compressive strength tends to be higher when the lower temperature of the freeze–thaw cycle drops. At room temperature, the addition of basalt fibers makes the internal pores of the concrete denser, and the fibers also effectively inhibit the lateral expansion of the concrete, retarding damage and thus increasing the compressive strength of the concrete. At lower temperatures, the concrete is more severely damaged internally, but the fibers are still effective, although their effectiveness is much reduced, and the compressive strength does not rise as significantly at this time. Under freeze–thaw cycles of −150 °C and −196 °C, 0.1% and 0.2% fibers may still be agglomerated, but the compressive strength still increases. This is because during the freeze–thaw cycles at −150 °C and −196 °C, some of the fiber bundles lose the ability to bond with the colloid, but when the fiber dosing reaches 0.2% or even 0.3%, there are enough fiber bundles to bear the load and inhibit the internal cracking by connecting with the matrix, thus reducing the damage to the internal structure of the concrete during ultra-low-temperature freeze–thaw cycles.

From Figure 7, it can be seen that under freeze–thaw cycles from room temperature to −150 °C, with an increase in freeze–thaw cycles, the compressive strength of BFRC concrete shows a slow decline and then a rapid decline trend at 0.2% admixture. The compressive strength of BFRC decreased by 7.16 MPa when the number of freezing and thawing cycles increased from 0 to 3, showing a decrease percentage of about 22.2%. Additionally, compressive strength further decreased by 12.22 MPa when the number of freezing and thawing cycles increased from 3 to 6, showing a decrease percentage of about 26.16%. This indicates that as the number of freeze−–thaw cycles increases, the evolution of the internal structural deterioration of BFRC is accelerated, resulting in the loss of bonding properties at the fiber–cement–aggregate interface.

### 3.2. Uniaxial Compression Deformation Test

To ensure the accuracy of the measured data, three strain gauges of 100 mm in length were glued to different sides with a resistance value of 120 ± 2 Ω and a sensitivity factor of 2.0 ± 1%. The arrangement of the strain gauges is shown in Figure 8.

The load cell, strain gauge, and displacement meter were connected to a DH3816N static strain collector produced by Jiangsu Donghua Testing Instruments Co., Ltd., (Taizhou, China) as is shown in Figure 9.

The initial displacement loading rate of the all-purpose press was set to 0.2 mm/min, and the loading rate was slowed down to 0.1 mm/min after loading to 70−90% of the predicted damage load. Compared with the strain of the rising segment, the strain of the falling segment is more difficult to measure directly via the strain gauges. Therefore, the strain in the falling segment was measured by a strain gauge combined with a displacement meter. The strain value after reaching the peak stress was calculated as follows:(1)$$\varepsilon =\varepsilon_{1}+\frac{\left(l-l_{1}\right)}{l_{0}}$$

In the formula:

ε—Strain after the peak load is attained.

$\varepsilon_{1}$—Strain corresponding to peak stress.

l—Displacement meter value after peak stress.

$l_{1}$—Displacement gauge value when peak stress is reached.

$l_{0}$—Longitudinal length of the specimen.

The main test data of the uniaxial compression deformation of the BFRC specimens by fiber admixture, temperature gradient, and number of freeze–thaw cycles are shown in Table 4. To effectively reduce the test error, the mean values of the three basic data for each test group were used as the final data.

The damage characteristics of the specimen under uniaxial compression with different fiber contents are shown in Figure 10.

The effect of the fiber admixture on the BFRC stress–strain curve in Figure 11 shows the stress–strain curves of concrete specimens with different fiber volume fractions. The initial curve reflecting a concave trend is due to the accumulation of damage inside the concrete specimens after freezing and thawing. The peak strain increased as the fiber volume fraction increased, which indicates that basalt fibers could effectively improve the ductile deformation capacity of the specimens after cryogenic freeze–thaw cycles. Compared with plain concrete, the decreasing trend of the curves of BFRC with the basalt fiber volume fraction of 0.1% and 0.2% were significantly slower after reaching the peak load. Additionally, the stress appeared to increase even further with strain, indicating that the concrete with basalt fibers exhibits obvious plasticity characteristics. Although the specimens were subjected to structural damage caused by freezing and thawing at temperatures ranging from room temperature to −150 °C, the appropriate amount of fiber was still able to resist the external load through friction and bonding with the matrix, prompting the concrete to change from brittle damage to plastic damage. When the fiber volume fraction increased to 0.3%, the plastic state of BFRC was inhibited to some extent, and the peak load and strain were reduced because the excessive addition of fibers presented a partial agglomeration, leading to local defects in the concrete. Additionally, the ultra-low-temperature environment caused further damage to the defect area.

The basalt fiber volume fraction of 0.2% and three cryogenic freeze–thaw cycles in Figure 12 shows that the stress in the elastic phase of BFRC at room temperature increases steadily with an increase in strain, and the curve in this phase shows an “upward convex” pattern. After reaching the peak load, the stress decreases steadily with an increase in strain, which indicates that the fiber bundles inside BFRC form a three-dimensional support structure by frictional bonding with matrixes. This indicates that the fiber bundles inside the BFRC form a three-dimensional support structure through frictional bonding with the cementitious matrixes, which effectively increases the peak stress and strain. When the temperature gradient is from room temperature to −80 °C, −150 °C, and −196 °C, the rising section of the stress–strain curve becomes more and more gentle, and the difference between the rising sections of the gradient from room temperature to −150 °C and from room temperature to −196 °C are smaller and stabilize. After BFRC is subjected to freeze–thaw cycles, the process of the freezing and thawing of internal water has an “extrusion effect” on the matrix, which results in the accumulation of small internal cracks, and the effect will be fully expanded in the cryogenic freeze–thaw environment, thus causing more damage to the subsequent axial compression deformation of BFRC. In addition, due to the strong resistance of basalt fibers to ultra-low temperatures, they can still produce viscous friction with a concrete matrix in better condition. Therefore, the internal structural damage mainly appears at the cementitious–aggregate interface, which is reflected by the deepening and stabilization of the internal damage. In the decreasing section of the stress–strain curve, BFRC still has good ductility characteristics under freeze–thaw cycles from room temperature to −80 °C and from room temperature to −150 °C. However, the stress–strain curve of BFRC under a freeze–thaw cycle from room temperature to −196 °C shows a rapid decrease, which indicates that the damage caused by a freeze–thaw cycle from room temperature to −196 °C is more serious and accelerates the ductility deterioration significantly.

As is shown in Figure 13, it can be observed that compared with 0 and 3 freeze–thaw cycles, the stress–strain curve of BFRC in the case of 6 freeze–thaw cycles showed an obvious “concave” phenomenon at the beginning, and then the curve accelerated and rose again. However, the peak stress and strain were greatly reduced, and the descending part dropped sharply with a larger degree of damage. This indicates that the internal structural damage of BFRC accumulates significantly as the number of freeze–thaw cycles increases. Additionally, the internal pores continue to develop to generate a large number of microcracks, which are squeezed and start to be closed under uniaxial compression. This is reflected in the curve characteristic of an obvious “concave”. The above situation aggravates the degradation of the cement aggregate fiber bundle system, which leads to damage to the toughness of the BFRC and a significant decrease in the load-bearing capacity and brittleness.

### 3.3. Toughness Index

To further investigate the toughness property of the BFRC after experiencing freeze–thaw cycles, the area under the stress–strain curve is one index that reflects material toughness. According to CECS38−2004, the toughness index can be used to assess the axial compression toughness of basalt fiber concrete. The calculation formula is as follows:(2)$$R_{e,1.0}=\frac{W_{1.0}}{F_{max}\times l_{0}\times 1.0\%}$$

In the formula:

$R_{e,1.0}$—Indicators of energy absorbed by the specimen.

$W_{1.0}$—Compression power (N·mm).

$F_{max}$—Peak load (N).

$l_{0}$—Axial length of test piece for compressive deformation measurement (mm), here it is 50 mm.

When the axial length of the test piece is certain, the larger the area enclosed between the stress–strain curve and the horizontal axis, the larger the toughness index; that is, the more energy the material absorbs in the process of loading. According to the stress–strain curve, the toughness index of each specimen undergoing different ultra-low-temperature cycles was calculated.

As can be seen from Figure 14, after the freeze–thaw cycles from room temperature to −150 °C, compared with plain concrete, the toughness index of BFRC with a fiber volume of 0.1% increased by 13.06%. When the fiber volume increased to 0.2%, the BFRC toughness index increased by 11.28%. When the fiber volume increased to 0.3%, the toughness index decreased by 13.40%. The increase in the uneven distribution of fibers inside the concrete due to the addition of excessive fibers leads to structural weaknesses and reduces its energy absorption effect after exposure to ultra-low temperatures.

The toughness index of BFRC showed accelerated decay as the temperature gradient of the ultra-low-temperature cycles increased. Compared with the toughness index of specimens experiencing freeze–thaw cycles from 20 °C to −80 °C, the toughness index of specimens experiencing freeze–thaw cycles from 20 °C to −150 °C and −196 °C decreased by 1.83% and 30.21%, respectively. This indicates that the appropriate amount of BFRC can effectively suppress the material toughness decay caused by the ultra-low temperature in the freeze–thaw environment from room temperature to −150 °C. However, under the influence of freeze–thaw cycles from room temperature to −196 °C, the internal damage to BFRC becomes more serious and the deterioration trend is increased.

### 3.4. Splitting Tensile Strength 

The splitting tensile damage of the specimens is shown in Figure 15. It can be seen that as the basalt fiber volume increased from 0 to 0.3%, the specimen crack length and width decreased significantly.

The concrete tensile strength is much lower than the compressive strength. BFRC can form a more effective three-dimensional disordered system when mixed with basalt fibers compared with ordinary concrete. When concrete experiences ultra-low-temperature cycles, it continuously forms accumulated damage. High bonding friction forms between the BFRP and the cementitious matrix due to the good tensile strength of basalt fibers under ultra-low temperatures. This limits the dispersion of aggregate and cementitious matrix after ultra-low temperature and effectively inhibits the expansion of transverse cracking under the splitting force. Thus, with an increase in basalt fiber admixture, the concrete crack section becomes more and more curved under splitting damage. The degree of crack development inhibition is increasing, and the concrete plasticity property is strengthened.

The relationship between the splitting tensile strength and fiber volume of BFRC under different freeze–thaw cycles with ultra-low temperatures of 20 °C, −80 °C, −150 °C, and −196 °C, respectively, are shown in Figure 16. Under the condition of room temperature of 20 °C, the splitting tensile strength of BFRC with a fiber volume fraction of 0.1%, 0.2%, and 0.3% increased by 11.2%, 6.34%, and 9.1%, respectively, compared with that of control specimens. This indicates that under a room temperature environment, the tensile strength of concrete was steadily improved by the incorporation of basalt fiber. Under the freeze–thaw condition from −80 °C to 20 °C, the splitting tensile strength with a fiber volume content of 0.1%, 0.2%, and 0.3% increased by 25.3%, 12.6%, and 7.0%, respectively, compared with that of the control specimens. This indicates that the admixture of a certain amount of basalt fiber can also improve the tensile strength under ultra-low-temperature freeze–thaw damage. Under the freeze–thaw conditions from room temperature to −150 °C, the splitting tensile strength of specimens with a fiber volume fraction of 0.1%, 0.2%, and 0.3% increased by 17.9%, 27.3%, and 6.1%, respectively, compared with the control specimens. Under the freeze–thaw conditions from room temperature to −196 °C, the percentages of tensile strength improvement with 0.1%, 0.2%, and 0.3% fiber volume ratio were 11.5%, 30.0%, and 1.2%, respectively, compared with the strength of control specimens. It can be seen that with a drop in the ultra-low temperature of freeze–thaw cycles, the deterioration of the tensile properties of plain concrete is more serious, and the incorporation of basalt fibers can effectively improve the tensile properties of concrete. However, concrete in a freeze–thaw environment with the ultra-low temperatures of −150 °C and −196 °C has a greater degree of damage due to there being more significant internal moisture freezing and swelling. Although high basalt fiber contents were dispersed, we still could not avoid the degradation of the concrete material. Performance improvement was no longer significant due to fiber incorporation.

As is shown in Figure 17, when the number of freeze–thaw cycles was smaller than 3, an appropriate amount of fiber incorporation can resist freeze–thaw damage. However, with an increasing number of iterations, the freeze–thaw damage effect was more serious, and the internal microcracks expanded and formed penetrations. This destroyed the bond property between the fiber and the concrete matrix and caused a partial loss of the splitting tensile strength of BFRC.

### 3.5. Flexural Strength

The three-point bending method was adopted here. The flexural testing machine and the loading method arrangement are shown in Figure 18. Load control mode was adopted, and the loading rate was 0.6 MPa/s.

To obtain the strength of the flexural specimens while utilizing three-point bending, the following equation was used:(3)$$f_{f}=\frac{3FL}{2bh^{2}}$$

In the formula:

$f_{f}$—BFRC flexural strength (MPa).

F—The maximum load value applied by the test (kN).

L—Longitudinal length of the specimen (m).

*b*, *h*—Breath and height of the specimen (m).

As is shown in Figure 19, the cracking openings of damaged specimens tended to be smaller when the fiber volume content was increased. During the loading process, the broken phenomenon of plane specimens happened very suddenly. On the other hand, For BFRC, the cracks developed more slowly for specimens with higher fiber content, indicating stronger ductility characteristics.

As can be seen from Figure 20, under the conditions of room temperature at 20 °C and freeze–thaw cycles from room temperature to −80 °C with plain concrete as the control specimen, when the amount of basalt fiber admixture was increased to 0.1%, the flexural strengths under the two temperature conditions increased by 5.9% and 11.0%, respectively. When the fiber admixture was increased to 0.2%, the flexural strengths increased by 20.6% and 26.7%, respectively. When the fiber admixture was increased to 0.3%, the flexural strengths increased by 7.1% and 6.4%, respectively. The concrete was less damaged internally under cryogenic freeze–thaw cycles from room temperature to −80 °C because a small amount of fibers can effectively inhibit the loss of flexural strength. However, as the temperature gradient continued to increase, the damage caused by cryogenic freeze–thaw cycles also increased, and the deterioration of the flexural properties increased significantly and eventually stabilized. With an increase in fiber admixture, the flexural strength of BFRC reflected a trend of first slow and then accelerated increase, and finally slowed down. Compared with plain concrete, BFRC can inhibit the development of microcracks in concrete under bending conditions through the bonding force generated by internal fiber bridging, thus improving its flexural strength.

Under the cryogenic freeze–thaw cycles from room temperature to −150 °C and from room temperature to −196 °C, the flexural strength increased by 14.9% and 11.5%, respectively, when the volume fraction of basalt fibers was increased to 0.1%. When the basalt fibers volume fraction was increased to 0.2%, the flexural strength increased by 21.0% and 6.5%, respectively. When the basalt fibers volume fraction was increased to 0.3%, the flexural strength increased by 7.1% and 3.5%, respectively. Overall, with an increase in the fiber volume fraction, the flexural strength of the BFRC showed a trend of rising and then slowing down. With a significant drop in the number of ultra-low-temperature cryogenic freeze–thaw cycles, the agglomeration phenomenon that exists in concrete rapidly expands in the cryogenic freezing and thawing state. Additionally, the cryogenic freezing and thawing cycles make the concrete’s internal interface of different materials form more serious damage. At this time, only through the interface enhancement effect brought about by high admixture can the damage be partially offset, which is reflected in the macro-level flexural strength increase.

From Figure 21, it can be seen that the decay of BFRC flexural strength slows down with an increasing number of cryogenic freeze–thaw cycles. When the number of cryogenic freeze–thaw cycles increases from 0 to 3, the flexural strength decreases by 22.5%. When it increases from 3 to 6, the flexural strength decreases by 33.4%. As the number of cryogenic freeze–thaw cycles increases, the degree of internal damage increases through the accumulation of cryogenic freeze-swelling effects. However, the decreasing trend of BFRC flexural strength is stabilized with a higher number of cryogenic cycles due to the appropriate amount of basalt fibers through the tensile effect between itself and the matrix concrete. Because the cracks on the damaged surface of the specimens without cryogenic cycles originate from the stress concentration effect, the cracking trend is clear. While the splitting phenomenon of coarse aggregate can be seen on the damaged surface of the specimens experiencing fewer cryogenic freeze–thaw cycles, this phenomenon is less obvious as the number of cryogenic freeze–thaw cycles increases and cracks or extensions of initial cracks appear in places such as the internal pores of concrete specimens and coarse aggregate surfaces, at which point the stress concentration phenomenon is relieved or even disappears [36,37].

### 3.6. SEM

SEM images of the fiber volume fraction of 0.1% and 0.2% and fiber-free BFRC samples at room temperature, −80 °C, −150 °C, and −196 °C are shown in Figure 22.

It can be seen in Figure 22a that there are large pores and cracks, the whole structure is relatively loose, and the natural strength is low. In Figure 22b, BF has been added, and thus the pores are tighter than before, and some fibrous C−S−H gels are formed in the pores [38], which indicates that the addition of a small amount of BF contributes to the early hydration reaction of concrete. In Figure 22c, it is obvious that the bond between concrete and BF is stronger than in the former, because more C−S−H gel is formed in the concrete, which further promotes the medium-term hydration reaction of concrete. Additionally, it can clearly be seen that more microporous structures are formed in the figure, and paste aggregate interfaces formed by calcium hydroxide (CH) crystals are rare [39], so the permeability is also lower. At this time, the tight concrete internal environment at this point also improves the compressive strength and splitting tensile strength of concrete.

In Figure 22b−h, concrete as a loose porous material can be seen with the fiber admixture fixed at 0.1% and 0.2% as the temperature constantly falls. At −150 °C and −196 °C, the destruction of the pore structure in the concrete is more obvious. The concrete material in the internal free water area adsorbed water, and as part of the small pore size capillary water freezing expansion, we can see the extrusion of the pore wall so that the concrete produces fine cracks. As the number of cryogenic freeze–thaw cycles and temperature gradient increase, the concrete mortar pore size gradually increase and the cracks continue to deepen. This further allows for the migration of external environmental moisture and more intense freezing and expansion, which provide more favorable conditions. This intensifies the process of concrete damage, and the splitting tensile strength of concrete is also gradually reduced in this process.

## 4. BFRC Mechanical Strength Fitting Analysis

Because the LNG tank is not opened and closed frequently in practice, the number of freeze–thaw cycles is locked at 3 in the BFRC fitting function.

### 4.1. BFRC Compressive Strength Fitting Function

The fitted functions of BFRC’s compressive strength with fiber admixture and a temperature of 20 °C, −80 °C, −150 °C, and −196 °C are shown in Table 5.

**Table 5 polymers-15-00196-t005:** Fitted functions of compressive strength with different fiber volume fractions and different temperature gradients.

Temperature Gradient	Fitting Function	
	Quadratic Functional Form	Trivial Functional Form
20 °C	(4) $$f_{cu}=\left(-2.60V_{f}^{2}+11.76V_{f}+33.05\right)\times f_{0}{}_{20{}^{\circ}C}$$ (FitR^2^ = 0.79)	(5) $$f_{cu}=\left(2.24V_{f}^{3}-19.36V_{f}^{2}+49.08V_{f}+9.58\right)\times f_{0}{}_{20{}^{\circ}C}$$ (FitR^2^ = 1)
−80 °C	(6) $$f_{cu}=\left(-3.08V_{f}^{2}+15.19V_{f}+26.0\right)\times f_{0}{}_{-80{}^{\circ}C}$$ (FitR^2^ = 0.85)	(7) $$f_{cu}=\left(1.93V_{f}^{3}-17.53V_{f}^{2}+47.36V_{f}+5.78\right)\times f_{0}{}_{-80{}^{\circ}C}$$ (FitR^2^ = 1)
−150 °C	(8) $$f_{cu}=\left(-0.61V_{f}^{2}+5.03V_{f}+27.31\right)\times f_{0}{}_{-150{}^{\circ}C}$$ (FitR^2^ = 0.82)	(9) $$f_{cu}=\left(-1.65V_{f}^{3}+11.75V_{f}^{2}-22.49V_{f}+44.6\right)\times f_{0}{}_{-150{}^{\circ}C}$$ (FitR^2^ = 1)
−196 °C	(10) $$f_{cu}=\left(0.72V_{f}^{2}-1.61V_{f}+31.56\right)\times f_{0}{}_{-196{}^{\circ}C}$$ (FitR^2^ = 1)	(11) $$f_{cu}=\left(-0.14V_{f}^{3}+1.77V_{f}^{2}-3.95V_{f}+33.03\right)\times f_{0}{}_{-196{}^{\circ}C}$$ (FitR^2^ = 1)

According to the above fitting function, a three-dimensional function relationship can be obtained with temperature as the *X*-axis, fiber volume fraction as the *Y*-axis, and compressive strength as the *Z*-axis:(12)$$f_{cu}=-139.125V_{f}{}^{2}+30.521V_{f}+0.041T-0.178V_{f}T+43.029\;\left({\mathrm{FitR}}^{2}=0.81\right)\;$$

In the formula:

T—Temperature gradient (20–196 °C).

$f_{cu}$—Compressive strength with different fiber admixture levels.

Vf—BF fiber volume fractions.

$f_{0}$—Compressive strength without fiber, specific temperature is reflected in Table 5.

The reason for the unsatisfactory fit of the three-dimensional functional relationship for compressive strength is that the specimens were not sensitive to fiber incorporation in the compressive test.

### 4.2. BFRC Splitting Strength Fitting Function

The fitted function curves of BFRC’s splitting tensile strength as a function of fiber volume fractions at 20 °C, −80 °C, −150 °C, and −196 °C are shown in Table 6.

**Table 6 polymers-15-00196-t006:** Fitted functions of splitting tensile strength with different fiber volume fractions and different temperature gradients.

Temperature Gradient	Fitting Function	
	Quadratic Functional Form	Trivial Functional Form
20 °C	(13) $$f_{ts}=\left(-0.32V_{f}^{2}+3.71V_{f}+3.84\right)\times f_{0}{}_{20{}^{\circ}C}$$ (FitR^2^ = 0.99)	(14) $$f_{ts}=\left(48.84V_{f}^{3}-22.30V_{f}^{2}+6.01V_{f}+3.84\right)\times f_{0}{}_{20\;{}^{\circ}C}$$ (FitR^2^ = 1)
−80 °C	(15) $$f_{ts}=\left(-11.63V_{f}^{2}+8.63V_{f}+3.05\right)\times f_{0}{}_{-80\;{}^{\circ}C}$$ (FitR^2^ = 1)	(16) $$f_{ts}=\left(20.17V_{f}^{3}-20.70V_{f}^{2}+9.58V_{f}+3.04\right)\times f_{0}{}_{-80\;{}^{\circ}C}$$ (FitR^2^ = 1)
−150 °C	(17) $$f_{ts}=\left(-5.42V_{f}^{2}+6.94V_{f}+2.47\right)\times f_{0}{}_{-150\;{}^{\circ}C}$$ (FitR^2^ = 0.97)	(18) $$f_{ts}=\left(-159.25V_{f}^{3}+66.25V_{f}^{2}-0.54V_{f}+2.52\right)\times f_{0}{}_{-150\;{}^{\circ}C}$$ (FitR^2^ = 1)
−196 °C	(19) $$f_{ts}=\left(-5.57V_{f}^{2}+5.48V_{f}+2.2\right)\times f_{0}{}_{-196\;{}^{\circ}C}$$ (FitR^2^ = 0.92)	(20) $$f_{ts}=\left(-190.04V_{f}^{3}+79.94V_{f}^{2}-8.36V_{f}+30.71\right)\times f_{0}{}_{-196\;{}^{\circ}C}$$ (FitR^2^ = 1)

According to the above fitting function, a three-dimensional functional relationship can be obtained with temperature as the *X*-axis, fiber volume fraction as the *Y*-axis, and splitting tensile strength as the *Z*-axis.
(21)$$f_{ts}=-5.813V_{f}{}^{2}+5.936V_{f}+0.004T-0.003V_{f}T+3.647\;{\mathrm{FitR}}^{2}=0.97$$

In the formula:

T—Temperature gradient (20–196 °C).

$f_{ts}$—Splitting tensile strength with different fiber admixture levels.

Vf—BF fiber volume fractions.

$f_{0}$—Splitting tensile strength without fibers, specific temperatures are reflected in Table 6.

### 4.3. BFRC Flexural Strength Fitting Function

The fitted equations for the variation of BFRC flexural strength with fiber admixture at 20 °C, −80 °C, −150 °C, and −196 °C are shown in Table 7.

**Table 7 polymers-15-00196-t007:** Fitting function of flexural strength with different fiber volume fractions and different temperature gradients.

Temperature Gradient	Fitting Function	
	Quadratic Functional Form	Trivial Functional Form
20 °C	(22) $$f_{f}=\left(9.92V_{f}^{2}+3.23V_{f}+4.03\right)\times f_{0}{}_{20{}^{\circ}C}$$ (FitR^2^ = 0.98)	(23) $$f_{f}=\left(-149.05V_{f}^{3}+77.0V_{f}^{2}-3.78V_{f}+4.08\right)\times f_{0}{}_{20{}^{\circ}C}$$ (FitR^2^ = 1)
−80 °C	(24) $$f_{f}=\left(-1.71V_{f}^{2}+6.85V_{f}+3.47\right)\times f_{0}{}_{-80{}^{\circ}C}$$ (FitR^2^ = 0.95)	(25) $$f_{f}=\left(-231.95V_{f}^{3}+102.67V_{f}^{2}+9.58V_{f}-4.0\right)\times f_{0}{}_{-80{}^{\circ}C}$$ (FitR^2^ = 1)
−150 °C	(26) $$f_{f}=\left(-8.09V_{f}^{2}+7.85V_{f}+2.60\right)\times f_{0}{}_{-150{}^{\circ}C}$$ (FitR^2^ = 0.99)	(27) $$f_{f}=\left(-90.35V_{f}^{3}+32.57V_{f}^{2}+3.6V_{f}+2.62\right)\times f_{0}{}_{-150{}^{\circ}C}$$ (FitR^2^ = 1)
−196 °C	(28) $$f_{f}=\left(-5.81V_{f}^{2}+3.96V_{f}+2.57\right)\times f_{0}{}_{-196{}^{\circ}C}$$ (FitR^2^ = 1)	(29) $$f_{f}=\left(17.93V_{f}^{3}-13.88V_{f}^{2}+4.8V_{f}+2.57\right)\times f_{0}{}_{-196{}^{\circ}C}$$ (FitR^2^ = 1)

According to the above fitting function, a three-dimensional functional relationship can be obtained with temperature as the *X*-axis, fiber volume fraction as the *Y*-axis, and flexural strength as the *Z*-axis.
(30)$$f_{f}=-1.563V_{f}{}^{2}+7.109V_{f}+0.016V_{f}T+3.821\;{\mathrm{FitR}}^{2}=0.96$$

In the formula:

T—Temperature gradient (20–196 °C)

$f_{f}$—Flexural strength with different fiber admixture levels.

Vf—BF fiber volume fractions.

$f_{0}$—Flexural strength in the absence of fibers, the specific temperature is reflected in Table 7.

## 5. Conclusions


(1)With an increase in the basalt fiber volume fraction, the crack width and number of BFRC specimens are significantly controlled when they are damaged, the brittle characteristics are significantly reduced when they are damaged, and the plasticity is strengthened.(2)Increasing the fiber volume fraction can significantly improve the mechanical strength of BFRC after ultra-low temperature cycles. With an increase in the fraction, the compressive strength of BFRC reaches its peak value when the fraction reaches 0.1% under the conditions of freezing and thawing cycles from room temperature to −80 °C. Under the normal temperature to −150 °C freeze–thaw cycle, the compressive strength reaches its peak value when the fraction reaches 0.2%, but under the normal temperature to −196 °C freeze–thaw cycle, the compressive strength reaches its peak value when the fraction reaches 0.3%. In addition, the splitting tensile strength and flexural strength reach their peak values when the content reaches 0.3% in each temperature gradient cryogenic cycle.(3)With an increase in the temperature gradient during the cryogenic cycle, the mechanical strength of BFRC after ultra-low temperature cycles is significantly reduced, and different fiber volume fractions affect this reduction trend. With the normal temperature specimen as the control, the compressive strength of BFRC under the conditions of freezing and thawing cycles from normal temperature to −80 °C, −150 °C and −196 °C decreases most significantly when the fraction of BFRC is 0.1%, and the maximum reduction is 17.1%. The splitting tensile strength decreased most sharply when the fraction was 0.1%, and the decrease amplitude was 40.9%. The flexural strength decreased most acutely when the fraction was 0.3%, and the maximum decrease was 44.62%.(4)An increased number of freeze–thaw cycles brings obvious damage to BFRC. Compared with zero freeze–thaw cycles of BFRC, the compressive strength, splitting tensile strength, and flexural strength of BFRC after six freeze–thaw cycles are decreased by 34.6%, 26.6%, and 36.03%, respectively. Under the influence of freeze–thaw times, the loss of flexural strength is the most obvious, followed by compressive strength, and the relative loss of splitting tensile strength is the least.(5)After ultra-low-temperature freezing and thawing cycles, the toughness index of BFRC first increases and then decreases with an increase in fiber fraction and reaches the peak toughness index value with 0.2% fiber fraction. With an increase in the temperature gradient and number of freeze–thaw cycles, the toughness index of BFRC generally decreases.(6)The compressive strength, splitting tensile strength, and flexural strength were analyzed by fitting functions for the four temperature points involved in the test to introduce the fitting functions of temperature and fiber volume fraction with the relevant strength in the range of 20 °C to −196 °C.(7)This study is only a preliminary exploratory study of the basic mechanical properties of basaltic fiber concrete in ultra-low-temperature freeze–thaw cycles, and there are still many scientific values in this field that are not yet known. In future studies, the interaction between ultra-low-temperature environment, fiber reinforced materials, and concrete should be studied through finite element simulations, which will provide more reference data and guidance for the design of fiber concrete structures in deep cold environments.


## Figures and Tables

**Figure 1 polymers-15-00196-f001:**
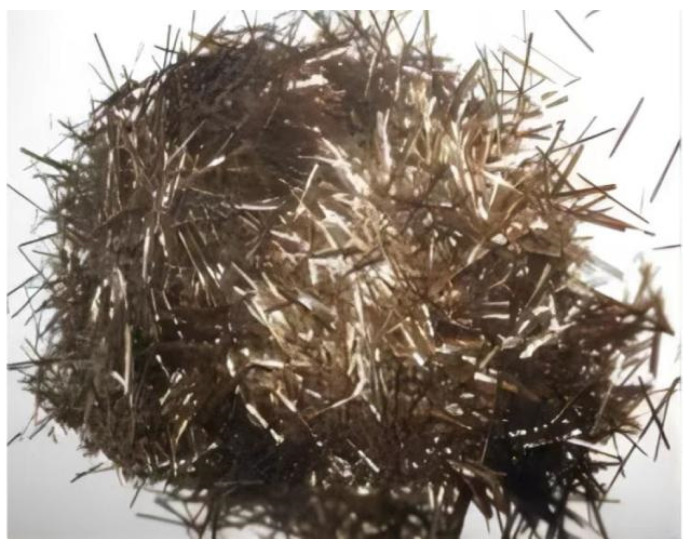
Basalt fiber samples.

**Figure 2 polymers-15-00196-f002:**
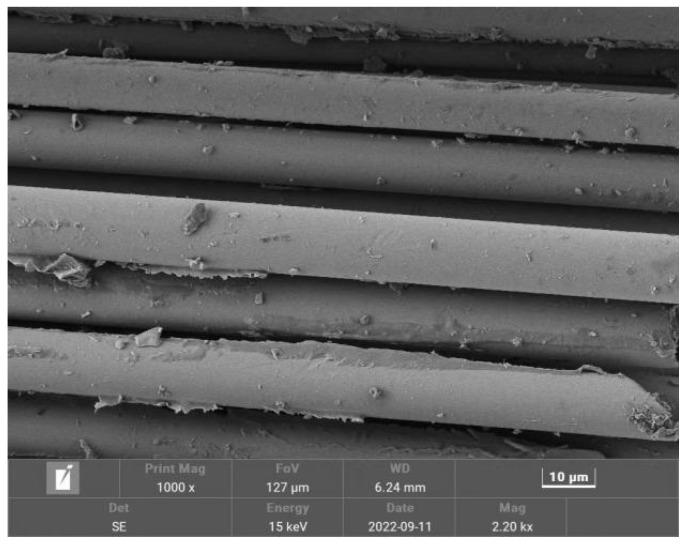
Characterization of basalt fiber morphology.

**Figure 3 polymers-15-00196-f003:**
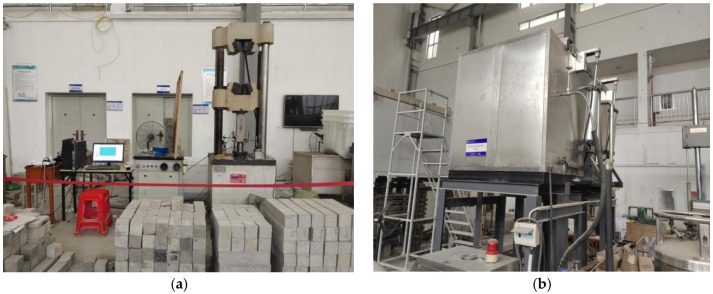
The main equipment of this experiment. (**a**) Mechanical strength test machine. (**b**) Cryogenic freeze–thaw cycle simulation test chamber.

**Figure 4 polymers-15-00196-f004:**
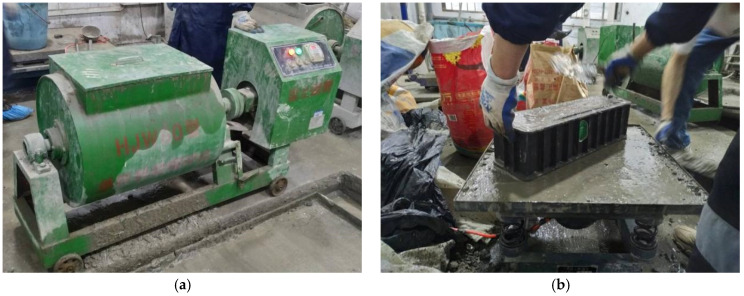
Test process. (**a**) Cement mixer. (**b**) Cement mortar pounding process.

**Figure 5 polymers-15-00196-f005:**
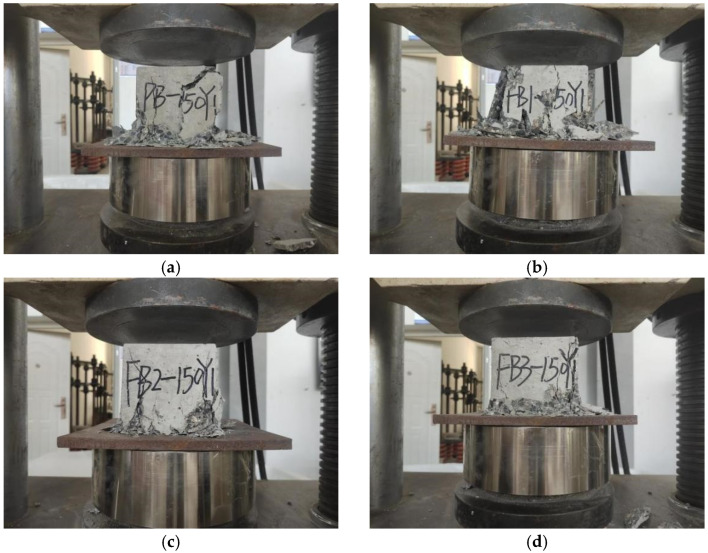
Specimen damage characteristics under pressure. (**a**) Specimen with 0% basalt fibers. (**b**) Specimen with 0.1% basalt fibers. (**c**) Specimen with 0.2% basalt fibers. (**d**) Specimen with 0.3% basalt fibers.

**Figure 6 polymers-15-00196-f006:**
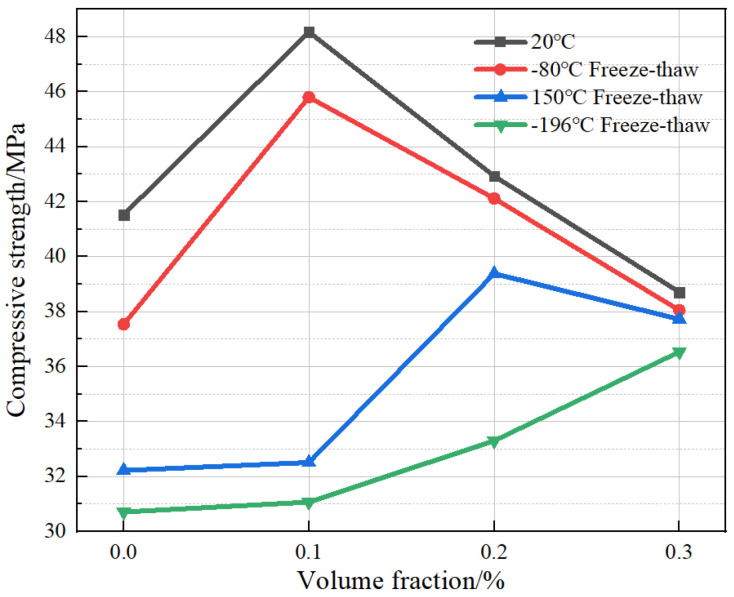
Effect of fiber content on the compressive strength of BFRC.

**Figure 7 polymers-15-00196-f007:**
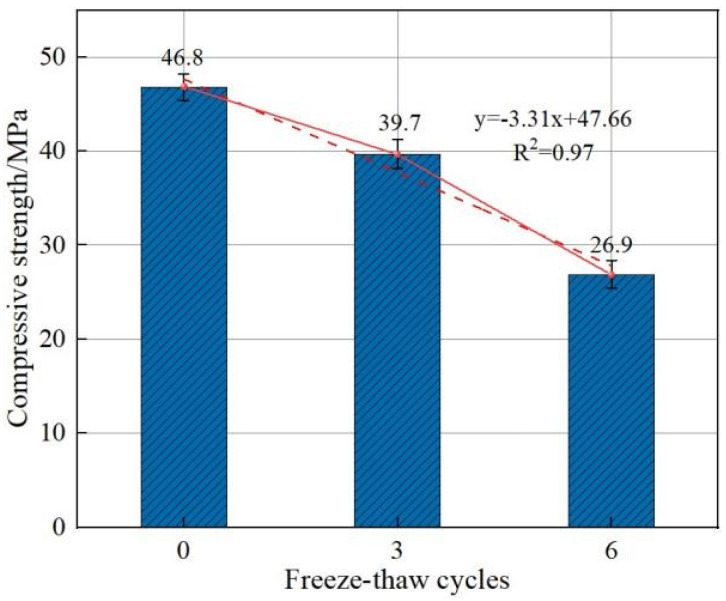
Effect of number of freeze–thaw cycles on the compressive strength of BFRC at ultra-low temperatures.

**Figure 8 polymers-15-00196-f008:**
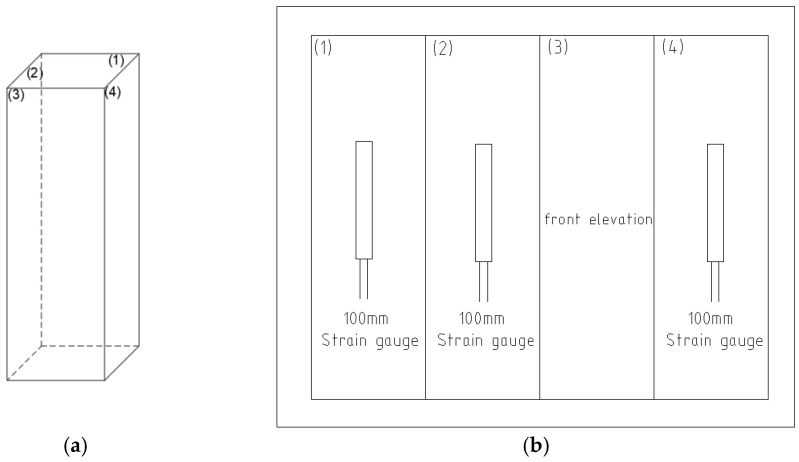
Strain gauge arrangement of BFRC specimen. (**a**) Concrete prisms. (**b**) Prismatic full-sided expansion.

**Figure 9 polymers-15-00196-f009:**
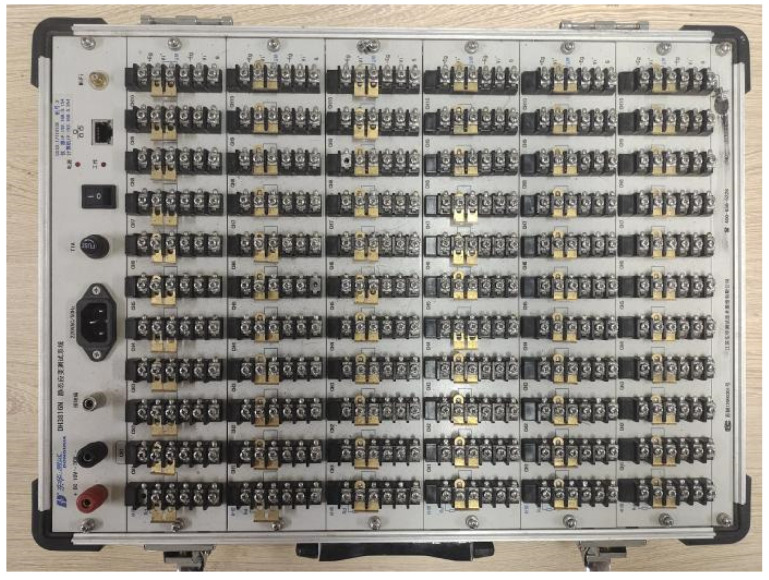
DH3816N static strain collector.

**Figure 10 polymers-15-00196-f010:**
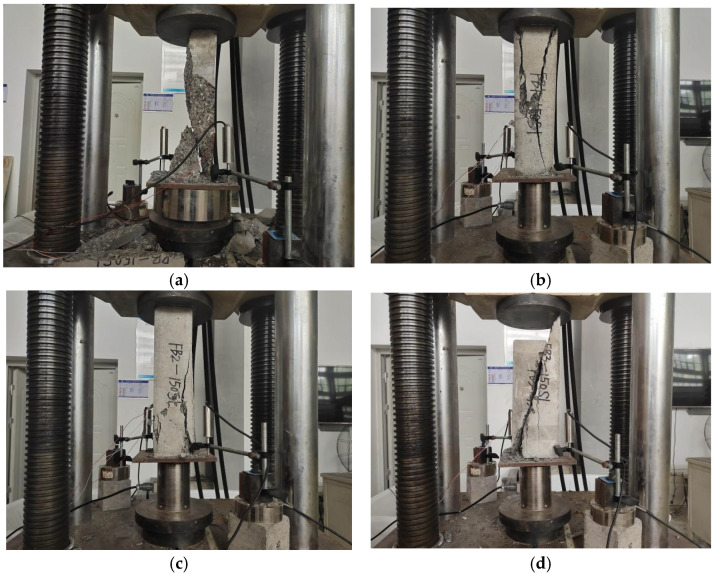
Damage characteristics of the specimen under uniaxial compression. (**a**) Volume fraction of 0%. (**b**) Volume fraction of 0.1%. (**c**) Volume fraction of 0.2%. (**d**) Volume fraction of 0.3%.

**Figure 11 polymers-15-00196-f011:**
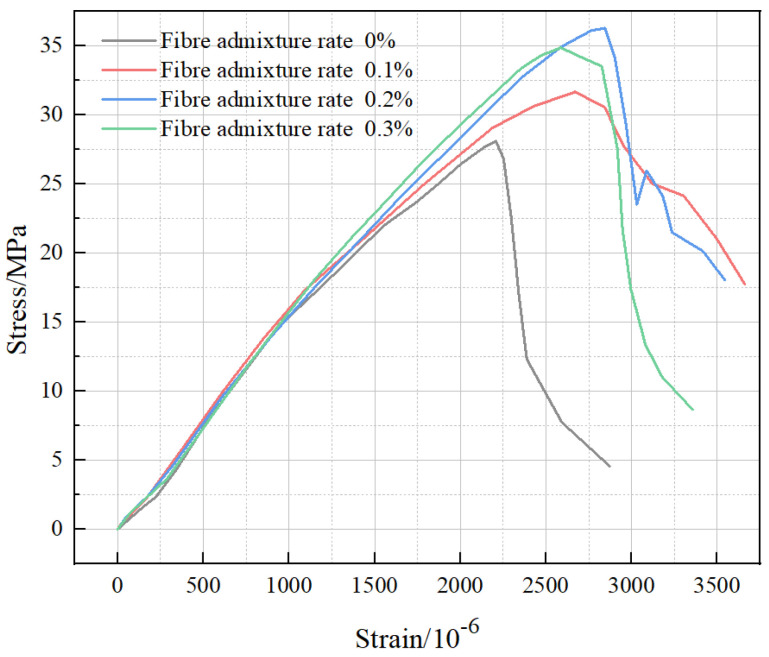
Effect of fiber admixture on BFRC stress–strain curves (freeze–thaw cycles from room temperature to −150 °C).

**Figure 12 polymers-15-00196-f012:**
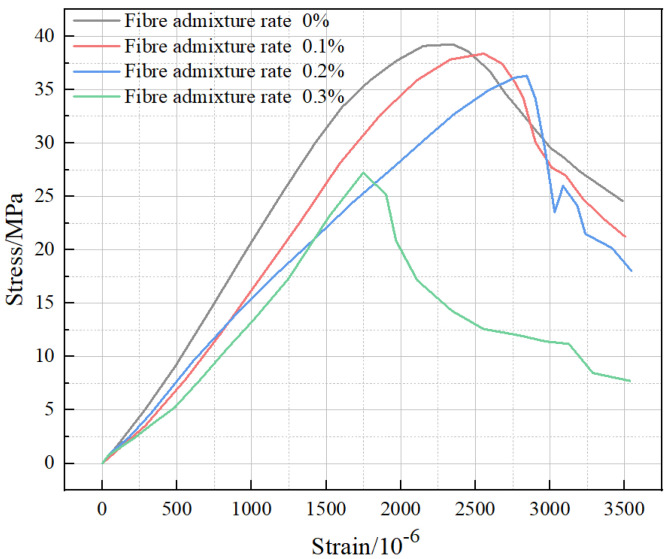
Effect of temperature gradient on BFRC stress–strain curve (freeze–thaw cycles from room temperature to −196 °C).

**Figure 13 polymers-15-00196-f013:**
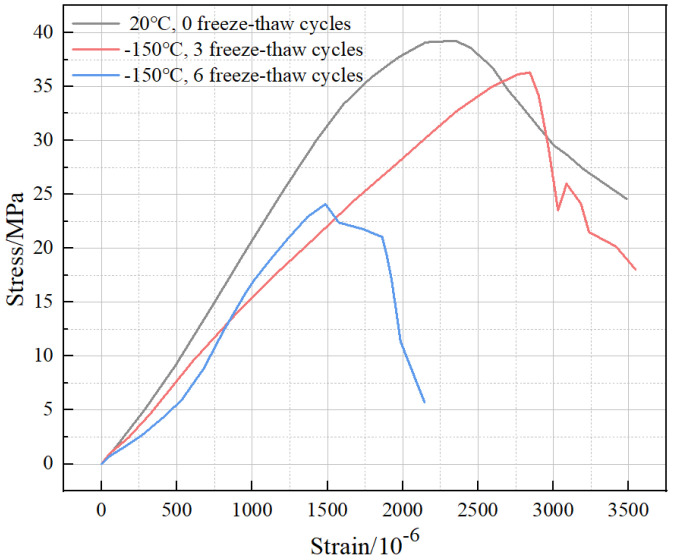
Effect of the number of freeze–thaw cycles on the stress–strain curve of BFRC (0.2% doping, 20 °C and −150 °C conditions).

**Figure 14 polymers-15-00196-f014:**
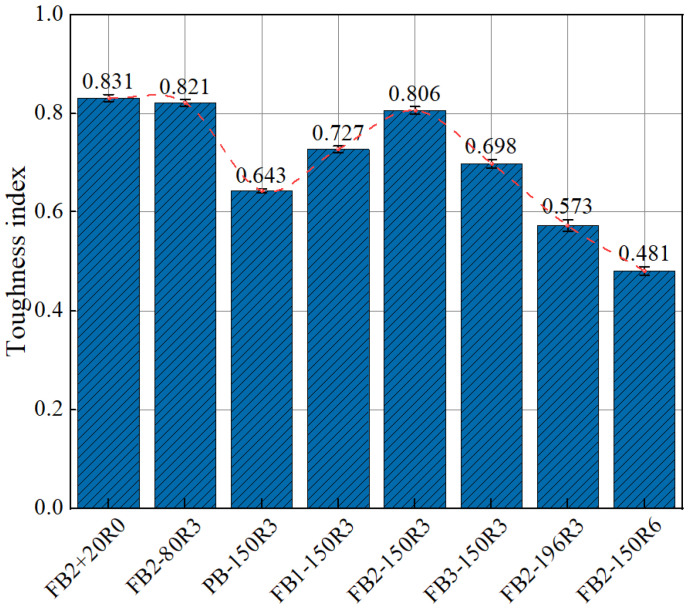
Statistics of BFRC toughness index under the influence of different factors.

**Figure 15 polymers-15-00196-f015:**
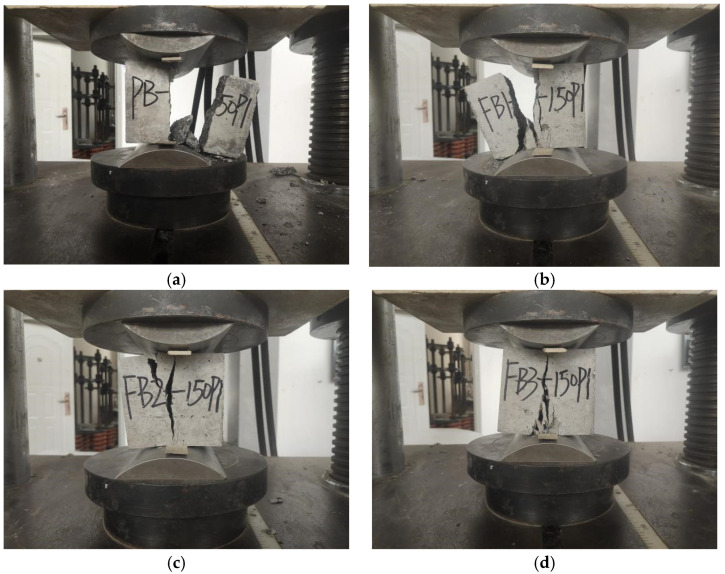
Splitting tensile damage characteristics of the specimen. (**a**) Volume fraction of 0%. (**b**) Volume fraction of 0.2%. (**c**) Volume fraction of 0.2%. (**d**) Volume fraction of 0.3%.

**Figure 16 polymers-15-00196-f016:**
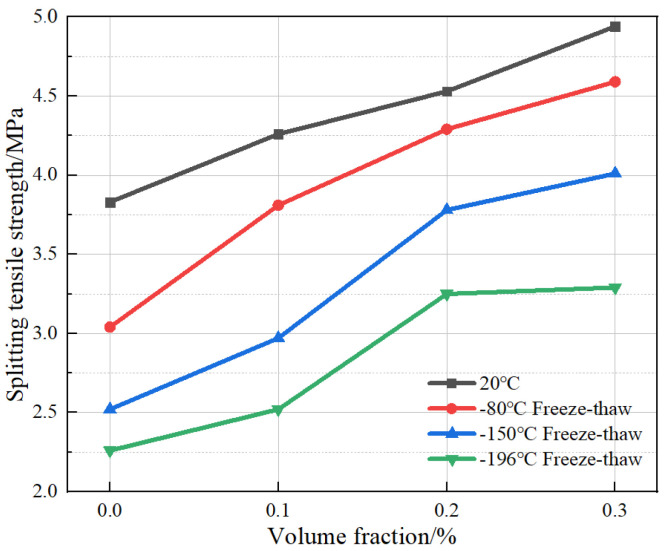
Effect of fiber volume on the splitting tensile strength of BFRC.

**Figure 17 polymers-15-00196-f017:**
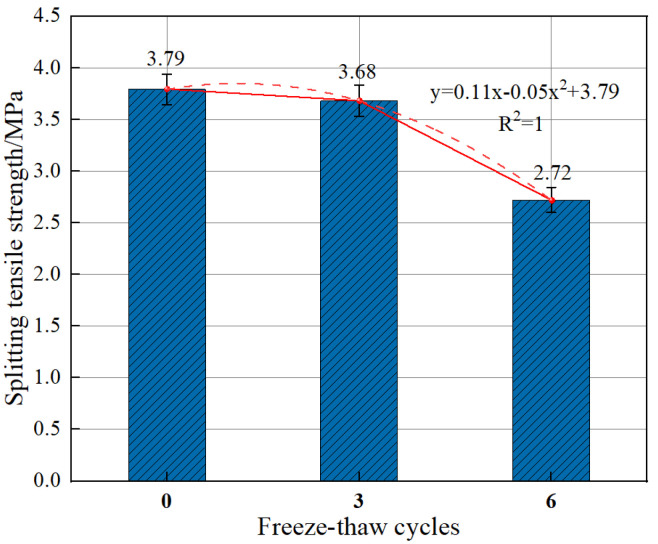
Effect of the number of freeze–thaw cycles on the splitting tensile strength of BFRC.

**Figure 18 polymers-15-00196-f018:**
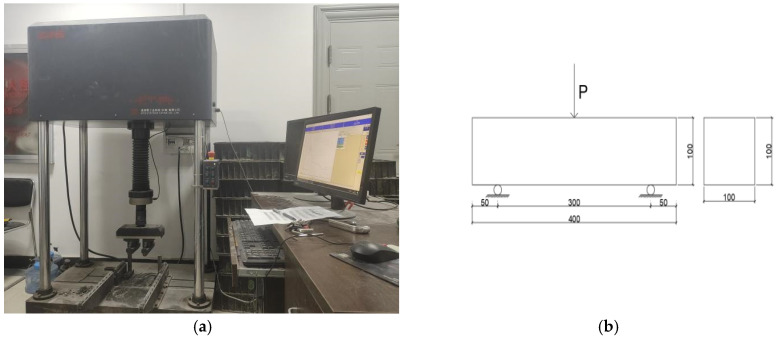
Flexural test equipment and loading device. (**a**) Test equipment. (**b**) Loading method arrangement.

**Figure 19 polymers-15-00196-f019:**
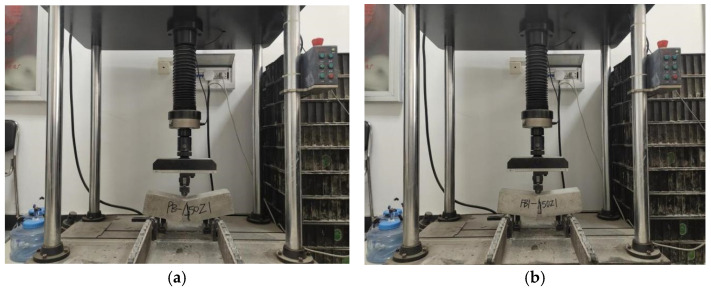
Flexural damage characteristics of the specimen. (**a**) Volume fraction of 0%. (**b**) Volume fraction of 0.1%. (**c**) Volume fraction of 0.2%. (**d**) Volume fraction of 0.3%.

**Figure 20 polymers-15-00196-f020:**
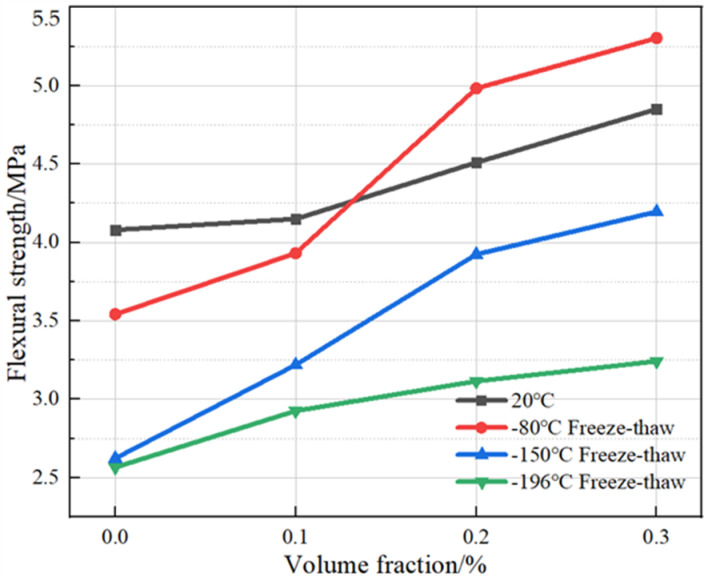
Effect of fiber volume on the flexural strength of BFRC.

**Figure 21 polymers-15-00196-f021:**
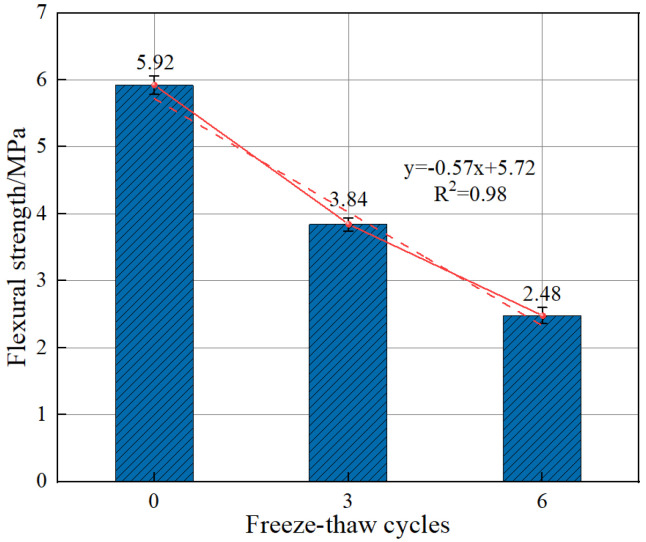
Effect of the number of freeze–thaw cycles on the flexural strength of BFRC.

**Figure 22 polymers-15-00196-f022:**
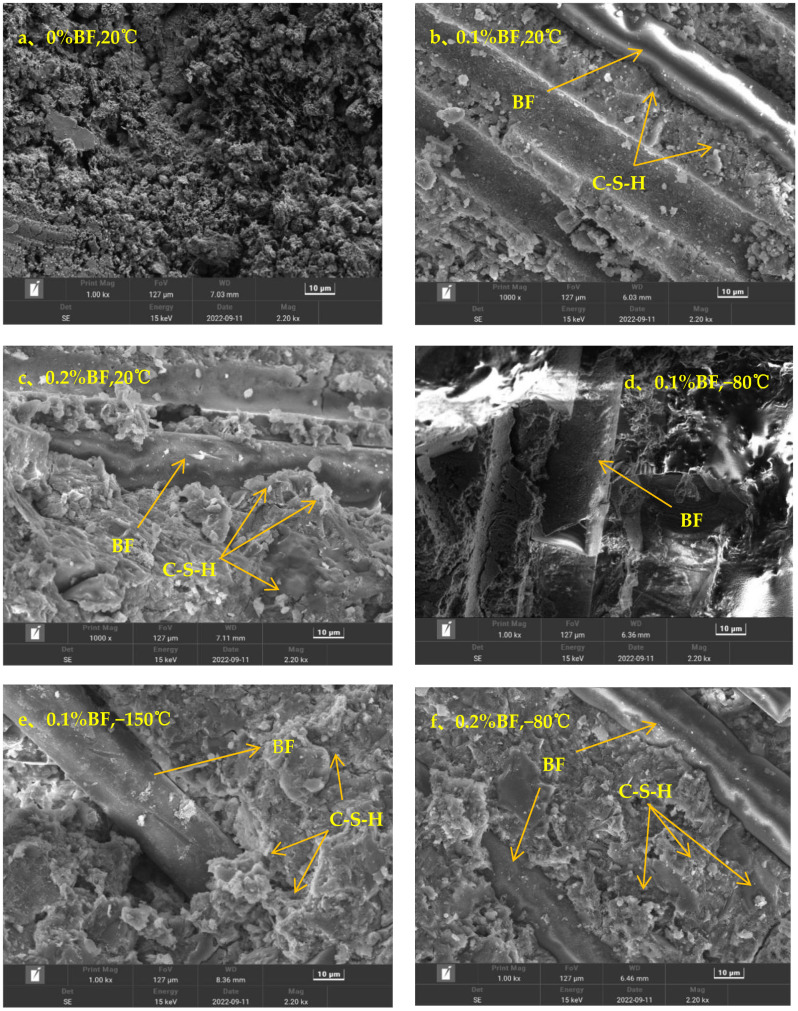
SEM image of basalt fiber concrete.

**Table 1 polymers-15-00196-t001:** Basalt fiber performance index.

Monofilament Diameter (µm)	Monofilament Length (mm)	Density (g/cm^3^)	Modulus of Elasticity (GPa)	Tensile Strength (MPa)	Elongation at Break (%)	Temperature Range (°C)	Thermal Conductivity (w/(m·K))
22	30	2.64	62	3836	3.0	−269–700 °C	0.03−0.038

**Table 2 polymers-15-00196-t002:** Concrete mix ratio (unit: kg/m^3^).

Water to Ash Ratio	Cement	Water	Sand	Gravel	Sand Rate
0.4	462	184	631	1127	36%

**Table 3 polymers-15-00196-t003:** Mix designs.

Specimen Number	Fiber Doping Rate (%)	Fiber Dose (kg/m^3^)	Water Cement Ratio	Temperature Gradient (°C)	Freeze–Thaw Cycles
PB+20−0	0	0	0.4	20	0
FB1+20−0	0.1	2.64			
FB2+20−0	0.2	5.28			
FB3+20−0	0.3	7.92			
PB−80−1	0	0	0.4	−80	3
FB1−80−1	0.1	2.64			
FB2−80−1	0.2	5.28			
FB3−80−1	0.3	7.92			
PB−150−1	0	0		−150	
FB1−150−1	0.1	2.64			
FB2−150−1	0.2	5.28			
FB3−150−1	0.3	7.92			
PB−196−1	0	0		−196	
FB1−196−1	0.1	2.64			
FB2−196−1	0.2	5.28			
FB3−196−1	0.3	7.92			
PB−80−1	0	0	0.4	−80	6
FB1−80−1	0.1	2.64			
FB2−80−1	0.2	5.28			
FB3−80−1	0.3	7.92			
PB−150−1	0	0		−150	
FB1−150−1	0.1	2.64			
FB2−150−1	0.2	5.28			
FB3−150−1	0.3	7.92			
PB−196−1	0	0		−196	
FB1−196−1	0.1	2.64			
FB2−196−1	0.2	5.28			
FB3−196−1	0.3	7.92			

**Table 4 polymers-15-00196-t004:** BFRC uniaxial compression peak strain and peak stress.

Specimen Number	Peak Strain/1 × 10^−6^	Peak Stress fc/MPa
FB2+20S0	2352	39.23
FB2−80S3	2257	38.38
PB−150S3	2002	26.46
FB1−150S3	2669	31.68
FB2−150S3	2844	36.30
FB3−150S3	2587	34.88
FB2−196S3	1747	27.21
FB2−150S6	1486	24.13

## Data Availability

The data presented in this study are available on request from the corresponding author upon reasonable request.

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
