# Peer review of "Experimental Study on Mechanical Properties of Basalt Fiber Concrete after Cryogenic Freeze−Thaw Cycles"

_polymers, 2022, doi:10.3390/polym15010196_

Round 1
Reviewer 1 Report
This article is good enough to provide solutions to the problem associated with LNG concrete storage tanks exposed to ultra-low temperature environments. Basalt fibers are successfully added to improve the mechanical properties of concrete for three different variables including fiber content, number of freeze-thaw cycles, and temperature cycling gradient.
However, the paper has the following shortcomings.
1. Authors quoted the names of different brands/Manufacturers for materials and equipment that should be avoided. Remain specific with their chemical and mechanical compositions.
2. Figure 1. And Figure 2. Can be placed in a single row.
3. Figure 1. Is not mentioned in the text.
4. No need to add figures of commonly used equipment (Figure 3 to Figure 8). these figure does not contribute to the research community. The author should mention the specifications of the equipment used (instead).
5. In Figure 10, the compressive strength of concrete was increased by the addition of basalt fiber from 0-0.1% for each freeze-thaw cycle. what will be the reason, to justify it?
6. Figure 10, What will be the reason for the increase in compressive strength for the cycle of-150◦C and -196◦C, however unevenly distributed and agglomeration of basalt fibers at 0.1 to 0.2% also exist there.
7. There must be a table for mix designs and freeze-thaw cycle.
Reviewer 2 Report
The article presents the current topic of experimental study on mechanical properties of basalt fiber concrete after cryogenic freeze-thaw cycles. While the work presents an interesting study, it requires a significant revision before it is recommended for publication.
1) In the abstract: the outcomes are too general and so please add some experimental results.
2) In the Introduction section: please add the research significance in the last paragraph.
3) In Figures 1 and 2 should be improved the quality of the images.
4) In Figures 5, 6, and 7 kindly add the images with samples of how to be conducting the experimental investigations like in figure 8.
5) Kindly add the compressive load versus deflection curves for all specimens.
6) Also add the flexural load versus deflection curves and explain their behaviors.
7) In the conclusion section: please add a future recommendation for your works.
Round 2
Reviewer 2 Report
As the authors have mitigated all comments in the revised manuscript. Therefore, I would like to recommend it for publication.